# Physiotherapy Intervention Improves Clinical Outcomes and Quality of Life in Elderly Patients with Osteoarthritis: A Prospective Cohort Study

**DOI:** 10.3390/ijerph22060966

**Published:** 2025-06-19

**Authors:** Jeel Moya-Salazar, Jordy R. Olortegui-Panaifo, Hans Contreras-Pulache, Eliane A. Goicochea-Palomino, Marx E. Morales-Martinez

**Affiliations:** 1Faculty of Medicine, Universidad Señor Sipan, Chiclayo 14002, Peru; 2Oficina de Normalización Previsional, YAYAQ Casa del Pensionista, Lima 51001, Peru; jordyolortegui@gmail.com; 3Faculty of Medicine, Universidad Norbert Wiener, Lima 15046, Peru; 4Faculty of Health Science, Universidad Tecnológica del Perú, Lima 15074, Peru; u19308295@utp.edu.pe; 5Department of Rehabilitation, Hospital Nacional Arzobispo Loayza, Lima 15077, Peru; mmoralesm@unfv.edu.pe; 6Faculty of Medical Technology, Universidad Nacional Federico Villareal, Lima 15001, Peru

**Keywords:** osteoarthritis, pain, elderly, physiotherapy, quality of life, Peru

## Abstract

Osteoarthritis is the most common disease among the elderly population and is expected to be one of the leading causes of physical disability worldwide. Our objective was to compare the effects of physiotherapeutic interventions versus pharmacological treatment on outcomes and quality of life in elderly patients with osteoarthritis. This cohort study was conducted on 119 elderly individuals aged 60 to 95 years (58.8% women) from the YUYAQ nursing home. Two groups were divided: the intervention group (58 individuals–48.7%) received a two-month physiotherapy program, and the control group (61–51.5%) received exclusive use of anti-inflammatories. Between the intervention and control groups, we observed significant improvements (all *p* < 0.001) regarding pain (93.1% vs. 60.75%), stiffness (94.8% vs. 62.3%), and functional capacity (96.6% vs. 68.9%). Additionally, the intervention group showed better quality of life than the control group (13.81 vs. 41.38, *p* < 0.001). Quality of life improvement and clinical outcomes in the treatment group significantly improved in all areas of osteoarthritis, primarily in hip, spine, and knee osteoarthritis (*p* < 0.001). In conclusion, the physiotherapy intervention improved pain, stiffness, functional capacity, and quality of life in elderly patients with osteoarthritis after two months of treatment. Transitioning from pharmacological treatment to physiotherapeutic treatment in patients with osteoarthritis may substantially improve quality of life and disease symptomatology, but long-term studies are needed.

## 1. Introduction

According to the World Health Organization (WHO), osteoarthritis or arthritis is the most common disease among older adults, affecting 528 million people worldwide by 2019 [1,2]. Especially in adults over 55 years old, the female population is more affected, with a higher incidence in the knees [1]. With the increase in the elderly population, osteoarthritis is expected to be one of the leading causes of physical disability worldwide. This is because this degenerative joint disease causes pain in the affected area, stiffness, and decreased functional capacity, ultimately affecting people’s quality of life [1,3].

Peru has undergone significant demographic aging, with the proportion of elderly citizens doubling from 5.7% in 1950 to 10.4% in 2018 [4]. This aging trend correlates with a rising prevalence of degenerative musculoskeletal conditions, particularly osteoarthritis. National epidemiological data reveal osteoarthritis as the most prevalent rheumatic disease, affecting 15.22% of the population in 2009, with a predilection for knee involvement (5.75%) and female predominance [5]. Updated statistics for 2016 demonstrate an incidence rate of 17.9 cases per 1000 inhabitants-year (12.3 in males vs. 23.7 in females), escalating to 72.6 cases per 1000 person-years among adults over 59 years [6], consistent with global patterns.

Osteoarthritis manifests primarily as movement-related daytime pain, though advanced stages may involve nocturnal or rest pain, reflecting central sensitization of nociceptive pathways that amplify pain perception [7,8]. Pain severity correlates strongly with clinical outcomes. Patients with severe pain exhibit greater joint involvement, higher hospitalization rates, and increased surgical demand [9]. Notably, 20–30% of total hip or knee arthroplasty recipients report suboptimal symptom relief or dissatisfaction one year postoperatively, highlighting limitations in surgical interventions as a definitive solution [10]. These findings emphasize the imperative for early identification of pain patterns and subtle structural changes during initial clinical evaluations. Such proactive assessment enables timely conservative management to decelerate disease progression and optimize long-term functional outcomes [10].

Current osteoarthritis management prioritizes symptom control, joint preservation, and functional improvement rather than curative approaches. Pharmacotherapy combines analgesics (e.g., paracetamol) and non-steroidal anti-inflammatory drugs (NSAIDs), though the latter require judicious use due to dose-dependent risks of gastrointestinal, cardiovascular, renal, and hepatic complications [11,12]. Emerging evidence advocates for NSAID administration at the lowest effective dose for minimal duration, particularly in older adults with comorbidities [12].

Physiotherapy plays a crucial role in this therapeutic approach, which includes the application of physical therapies. These include thermotherapy, which helps relax muscles and increase blood circulation in the affected area, thus reducing pain and stiffness [13], as well as manual therapy and exercises (stretching, balance, and resistance) to maintain mobility and prevent progressive loss of muscle strength [14]. Tailored therapeutic exercise demonstrates particular efficacy in slowing disease progression, though it requires careful movement selection to avoid excessive joint loading [15]. Comprehensive physiotherapy programs not only improve physical outcomes but also enhance rehabilitation adherence and patient satisfaction by addressing individual functional needs [15].

Although there is no standardized program implemented in various Peruvian health centers, research has shown that physiotherapy, especially muscle strengthening, significantly improves joint mobility and both dynamic and static balance in patients with osteoarthritis [16]. Additionally, it is essential to adapt exercise programs to the capabilities and individual needs of patients to achieve the best results [17]. For this reason, the objective of this study was to compare the effects of a physiotherapeutic intervention on pain, stiffness, functional capacity, and perceived quality of life. The hypothesis of the study posits that the group undergoing physical therapy would experience reduced joint pain and stiffness, as well as enhanced functional capacity and quality of life, in comparison to the control group that received pharmacological treatment.

## 2. Materials and Methods

### 2.1. Study Design, Population, and Inclusion Criteria

A cohort study was designed with an intervention group (physiotherapy treatment) and a control group (use of NSAIDs). The study population consisted of 119 elderly individuals diagnosed with osteoarthritis from the YUYAQ nursing home. YUYAQ is a retirement home where retirees are accommodated after their contributions to the Office of Pension Standardization (ONP) according to Peruvian legislation [18]. Non-probability sampling was used since all elderly residents of the nursing home were included, with the entire population as the unit of analysis. The research included all the elderly in YUYAQ; therefore, the entire available population was studied.

Inclusion criteria comprised older adults (≥60 years) of both genders residing at the YUQAY home who met the following: (1) provision of written informed consent; (2) absence of diagnosed psychiatric conditions (verified through medical records); and (3) preserved ambulatory capacity with functional independence in activities of daily living (ADLs), requiring no more than minimal assistance (e.g., walking aids). Cognitive eligibility was confirmed via baseline screening (e.g., Mini-Mental State Examination score ≥ 24) to ensure protocol adherence (data no showed). Exclusion criteria for both groups included cognitive deficit or psychological disorder, recent surgical history (surgeries in the knee, hip, and spine areas), visual, auditory, or physical disability, as well as incorrect questionnaire responses or failure to sign the informed consent form (Figure 1). Simple random sampling took place, and throughout the study, two patients—one from each group—withdrew due to travel commitments.

### 2.2. Intervention Group

The treatment group consisted of 58 elderly individuals from the YUYAQ program who received a minimum of one month of a series of yoga and physical activity workshops between January and July 2022. This intervention program was designed to promote active and healthy aging through dance, yoga, chi kung, cardio dance, and tai chi workshops [18]. The physiotherapy program was divided into three sessions per week, each lasting two and a half hours. These included dance (20 min), yoga (30 min), tai chi (30 min), and chi kung or cardio dance (30 min). A 10 min light warm-up and a 20 min post-intervention recovery session were included. The pain was measured with the numerical visual scale (NVS), and for this group, the average pain score was 7 ± 2 points.

### 2.3. Control Group

The control group comprised 61 elderly individuals diagnosed with osteoarthritis who maintained pharmacological treatment (NSAIDs). Three NSAID regimens were followed. The first involved naproxen 500 mg PO every 12 h, the second involved paracetamol 500 mg PO every 8 h, and the third included paracetamol 500 mg every 8 h with dexamethasone 20 mg every 24 h (for severe pain), both PO. Patients had good pharmacological adherence because no treatment discontinuations were observed during the study period. The average pain score was 5 ± 1 points.

### 2.4. Instruments

Two data collection instruments were used. The first was a record form for sociodemographic and clinical data, including age, occupation, marital status, education level, weight, height, Body Mass Index, nutritional status, comorbidities (hypertension, diabetes mellitus, others), affected joint, and medication use.

The second instrument was the “Western Ontario and McMaster Universities Osteoarthritis Index”—WOMAC questionnaire, designed in 1988 by Bellamy et al., to measure symptoms and perceived physical disability in individuals with hip or knee osteoarthritis [19]. Its metric properties have been demonstrated in numerous studies, showing sensitivity to change [20,21]. Its adaptation to Spanish was carried out in 1999 by Batlle-Gualda et al. [22], and it was validated in the Peruvian population the same year by Glave-Testino et al. [23] at the Universidad Nacional Mayor de San Marcos. The WOMAC questionnaire printed in Spanish was used for both study groups. The questionnaire has 24 items grouped into three sections, each with its own score: pain (5 items) from 0 to 20 points, stiffness (2 items) from 0 to 8 points, and functional capacity (17 items) from 0 to 68 points. A higher score indicates greater perceived disability by the patient and lower quality of life [24].

Another variable to be compared was quality of life, which will be obtained by summing the scores from the pain, stiffness, and functional capacity scales of the WOMAC. In this study, quality of life is considered a continuous quantitative variable, with intervals categorized as good quality of life (0 to 30 points), fair (31 to 60 points), and poor (61 to 96 points) [25].

### 2.5. Variables and Data Gathering

The study variable was perceived quality of life through the assessment of pain, stiffness, and functional capacity in patients with osteoarthritis. Initially, patients were recruited through study dissemination talks at the YUYAQ nursing home, selected, and randomly assigned to intervention and control groups. All participants signed a printed informed consent form before the start of the intervention. Physiotherapy intervention activities were scheduled according to the patients’ schedules and in coordination with the healthcare staff at the nursing home. The questionnaire was administered in person at the end of the physiotherapy intervention and follow-up period for each study group. An evaluation of the quality of filled forms was conducted to avoid errors in data collection. The caregivers, patients, and technical staff of YUYAQ had no information about the assigned groups.

### 2.6. Data Analysis

For processing the obtained information, a database was created in MS-Excel for Windows where study data were recorded using a standardized code. Descriptive analyses were performed using simple frequencies, percentages, means, and standard deviations. The Kolmogorov–Smirnov test for normality and paired *t*-test were conducted to demonstrate differences in clinical outcomes and quality of life among patients. A significance threshold of *p* < 0.05 and a 95% confidence interval were considered. Statistical processing and analysis of the results were conducted using the Statistical Package for the Social Sciences (SPSS) version 26.0 (IBMS, Armonk, NY, USA), BoxPlotR (Tyers and Rappsilber Labs, Berlin, Germany), and SankeyMATIC (Sankey library, D3. JS) [26].

### 2.7. Ethical Aspects

This study complied with the research guidelines of the Helsinki Declaration [27] and CIOMS [28]. Additionally, this research received approval from the YUYAQ Program of the Pensioners’ House (Report 01-2022) and was approved by the Ethics Committee of the Universidad Tecnológica del Perú (File 014-2022).

## 3. Results

Of the total participants, 70 (58.8%) were female, and 66 (55.5%) were in the age range of 60 to 69 years of age. The average BMI was 26.85, with 37 (31.1%) classified as overweight and 4 (3.4%) as obese. The age (average 68.8 ± 7.3, 95% CI: 66.9 to 70.7) of the control group did not differ from the age (average 71.3 ± 5.9, CI95% 69.7 to 72.8) of the treatment group (*p* = 0.113). However, differences (*p* < 0.007) were observed in the BMI of the control group (average 27.6 ± 3.2, 95% CI: 26.8 to 28.4) compared to the treatment group (average 26.1 ± 2.9, 95% CI: 25.3 to 26.8). Regarding the location of the pathology, 47 (39.49%) had osteoarthritis in the knee, 40 (33.61%) in the hip, and 32 (26.89%) in the spine. Additionally, 13 participants (10.9%) had more than one affected area by osteoarthritis, specifically 9 (69.2%) had two affected joints, and 4 (30.8%) had three joints affected. Furthermore, 58 participants (48.74%) engaged in physical activity, and 61 (51.26%) consumed medication (Table 1).

In addition, the total score of the WOMAC questionnaire in the treatment group was 13.81, indicating a high quality of life, whereas in the control group, a score of 41.38 was obtained, indicating a regular quality of life (Figure 2). Differences between the treatment and control groups were found in pain type (2.86 vs. 8.46), stiffness (0.89 vs. 3.77), functional capacity (10.02 vs. 29.15), and quality of life (13.81 vs. 41.38) (all *p* < 0.001).

Regarding symptoms and perception of physical disability, the majority of the elderly experienced mild pain. Specifically, 93.1% of the treatment group, engaged in physical activity, reported mild pain, followed by 6.9% with moderate pain. In contrast, in the control group, consuming medication, moderate pain was more common (60.75%), followed by mild pain (36.1%). Pain had an average score of 8.5 (±3.2) in the control group and a better score of 2.9 (±2.5) in the treatment group.

As for stiffness, the majority experienced mild stiffness (58.0%), predominantly in the intervention group (94.8%). Thus, 62.3% of the control group had moderate stiffness, followed by 23% with low stiffness. Stiffness had an average score of 3.8 (±1.6) in the control group and 0.9 (±0.9) in the treatment group.

Finally, most participants demonstrated adequate functional capacity (61.3%), with a notable 96.6% in the intervention group. In contrast, 68.9% of the control group experienced difficulties, while 27.9% maintained adequate functional capacity. It is important to note that this latter group was the only one with participants experiencing severe pain and stiffness (both 1.7%), as well as severe functional capacity difficulties (1.7%) (Table 2). On average, the control group had a final score of 29.1 (±10.2), and in those who received treatment, it was 10.0 (±6.9).

Regarding quality of life, scores ranged from 37.90 to 44.85 in the control group with an average of 41.4 (±13.8), and in the treatment group, they ranged from 11.31 to 16.31 with an average of 13.8 (±9.7). Moreover, most of the control group (73.8%) had a regular quality of life, followed by 23.0% with good quality of life, and it was the only group to have poor quality of life (two participants, 3.3%) (*p* < 0.04). In contrast, 96.6% of elderly individuals receiving physiotherapy treatment had a good quality of life, representing between 75 to 100% of those affected in one to three joints (100% of those with hip, spine, and two joint arthritis; 94.7% with knee arthritis, and 75% in three joints). The rest of the participants had a regular quality of life (5.3% had knee arthritis, and 25% in three joints) (*p* < 0.001). Thus, a significant difference was found in both groups, where those receiving physical therapy generally had a better quality of life than the control group (*p* < 0.001) (Figure 3).

## 4. Discussion

This study evaluated the comparative effectiveness of physiotherapy versus pharmacological management in enhancing functional outcomes and quality of life among older adults with osteoarthritis. Results demonstrated that participants receiving a multimodal physiotherapy program achieved statistically significant improvements in quality-of-life metrics compared to the pharmacotherapy-only control group. The physiotherapy cohort exhibited clinically meaningful reductions in joint pain intensity and stiffness, alongside enhanced functional mobility in activities of daily living.

### 4.1. Strengths

One strength of the study lies in the literature review concerning the assessment of the impact of physiotherapy intervention on patients with osteoarthritis, where it was found that the WOMAC questionnaire is particularly useful for measuring perceived quality of life in this population [23,29,30]. Additionally, previous studies highlight its ease of application and suggest its integration into routine clinical practice to evaluate the effectiveness of various therapeutic options [30], as assessing functional capacity and pain are crucial for establishing concrete projections regarding the functional dependency of individuals with severe osteoarthritis. Furthermore, the results of this research can provide valuable evidence for physiotherapy and for planning health strategies at the national level [31].

### 4.2. Main Findings

Our results show that compared to the control group treated with medication, the group treated with the physiotherapy program had low scores on the WOMAC questionnaire (pain: 8.5 vs. 2.9, stiffness: 29.1 vs. 0.9, and functional capacity: 29.1 vs. 10.0), which shows lower physical disability and thus a better quality of life (41.4 vs. 13.8). This is confirmed by studies that evaluated the before and after effects of physiotherapy intervention in patients with osteoarthritis. In elderly Cubans treated with infrared heat application, massages, and exercise, a greater than 50% improvement was observed after treatment. Specifically, the WOMAC showed that pain improved by 19.4% (pre-treatment: 15.92, post-treatment: 3.88), stiffness by 38% (pre-treatment: 6.8, post-treatment: 3.04), and functional capacity by 45% (pre-treatment: 62.8, post-treatment: 30.8) [30]. This was also seen in India, where there was significant improvement after 30 days of conventional physiotherapy treatment, including transcutaneous electrical nerve stimulation (TENS) and a routine of isometric and strengthening exercises. The pain score decreased from 13.6 to 7.3, stiffness decreased from 4.52 to 2.05, and functional capacity decreased from 48.5 to 19.17 [32]. In both cases, it is emphasized that the physiotherapy program was and should be practical, economical, and tailored to the goals of each patient [30,32].

Our results showed that the knee was the most affected region by osteoarthritis, a conclusion consistent with previous research at the national and international levels [1,5]. Additionally, patients undergoing physiotherapy treatment exhibited significantly higher scores in all sections of the WOMAC questionnaire, and a notable 94.7% reported good quality of life. These results are supported by two specific studies focusing on grade II knee osteoarthritis patients. One of these studies was conducted in a Peruvian military hospital, where a significant impact of the physiotherapy program on quality of life was observed. Initially, the average total score was 41 (indicating regular quality of life), but after treatment, this figure decreased significantly to 27 points (indicating high quality of life) [31]. A similar finding was recorded in Mexico, where the initial score decreased from 33.70 to 10.70 after physiotherapy intervention [33].

However, it is important to note that these investigations did not address the emotional state of the patients, a factor that can also have a significant impact on their quality of life [33]. Studies emphasize that sometimes pharmacological treatment fails to completely relieve symptoms and may have side effects that affect other systems or interact with other medications the patient may be taking. For this reason, the importance of initiating physiotherapy treatment as soon as osteoarthritis is diagnosed and simultaneously with pharmacological treatment is highlighted, as the combination of both approaches is more effective, especially in pain management [30].

On the other hand, research has been found that does not agree with our marked improvement after physiotherapy treatment. In a Turkish population, after a 12-session treatment over six weeks that incorporated physical agents such as hot compresses, ultrasound, TENS, and an exercise program, the initial quality of life score was 59.04 ± 21.49 and only improved to 52.28 ± 19.54 after intervention. However, the authors noted a significant improvement in functional status (*p* < 0.05) and highlighted that the combination of several physical agents often produces more satisfactory results at the end of treatment [34]. This evidence indicates that the final quality of life score of the participants was low despite the applied intervention. It is possible that the quality of life of these patients was affected by contextual factors related to the COVID-19 pandemic, such as lockdowns and lifestyle changes that may affect mental health [35]. During the lockdown, exercises have also been reported to be fundamental and, to some extent, have contributed to reducing mental health problems in older adults [36]. Therefore, they could also delay the progression of the disease as much as possible [16].

### 4.3. Limitations

First, due to restrictions caused by the COVID-19 pandemic, it was difficult for older adults to respond to virtual surveys, so in-person surveys were conducted where researchers supervised the completion of the questionnaire by the elderly participants. Both YUYAQ workers and family members supported some patients during the questionnaire [37]. However, when evaluating patients from different locations (municipalities of Independencia, San Juan de Lurigancho, and San Juan de Miraflores), there may have been biases among the evaluated populations since the social and economic characteristics vary [38].

Second, it is important to consider that other factors may contribute to the beneficial effects of the intervention. For example, in Peru, an active aging program is recommended by the Ministry of Health for public, private, and non-governmental institutions [39]. Our participants participated in these programs at home or before entering YUYAQ through regular physical activity workshops [18]. Some studies suggest that implementing yoga programs can improve the effectiveness of conventional physiotherapy programs in these patients [32]. Since these data are not available, they may have influenced the final study assessment. Further studies are needed to assess these factors in the outcomes of physiotherapy activities.

Third, we did not take into account whether participants had received other additional treatments, such as acupuncture, before the start of the intervention. It is important to consider that previous therapies that reduce pain and improve functional status could influence our results [34].

Finally, the emotional state of the patients was not evaluated, despite its impact on quality of life, even in those with good physical condition [33]. It is essential to consider this aspect in future research to better understand the comprehensive impact of interventions on participants’ health and well-being.

## 5. Conclusions

Elderly individuals with osteoarthritis who engage in physical activities and receive physiotherapy counseling show less stiffness, greater endurance, and functional capacity than elderly individuals undergoing anti-inflammatory treatment. Physiotherapy intervention, by improving clinical outcomes, also impacts quality of life, enhancing it two months after the intervention. Further research is necessary to understand the long-term effects as well as the most efficient types of therapies with better impact on areas affected by osteoarthritis and other chronic diseases.

## Figures and Tables

**Figure 1 ijerph-22-00966-f001:**
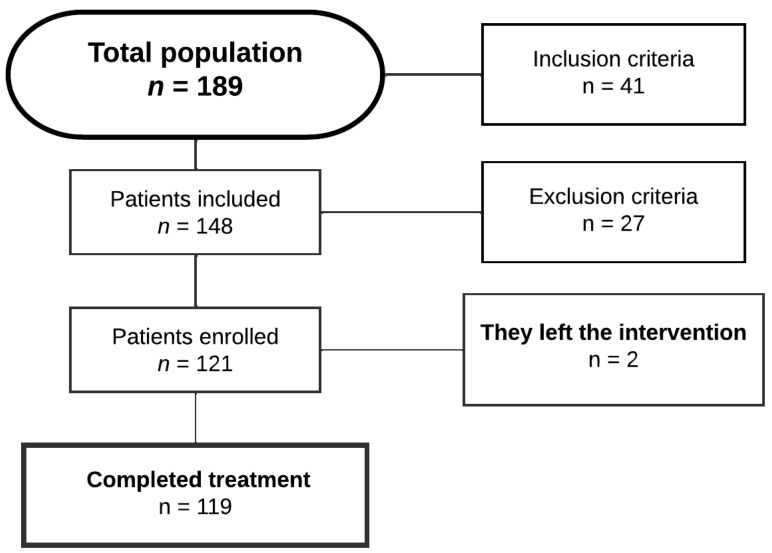
Flowchart for the selection of elderly individuals with osteoarthritis.

**Figure 2 ijerph-22-00966-f002:**
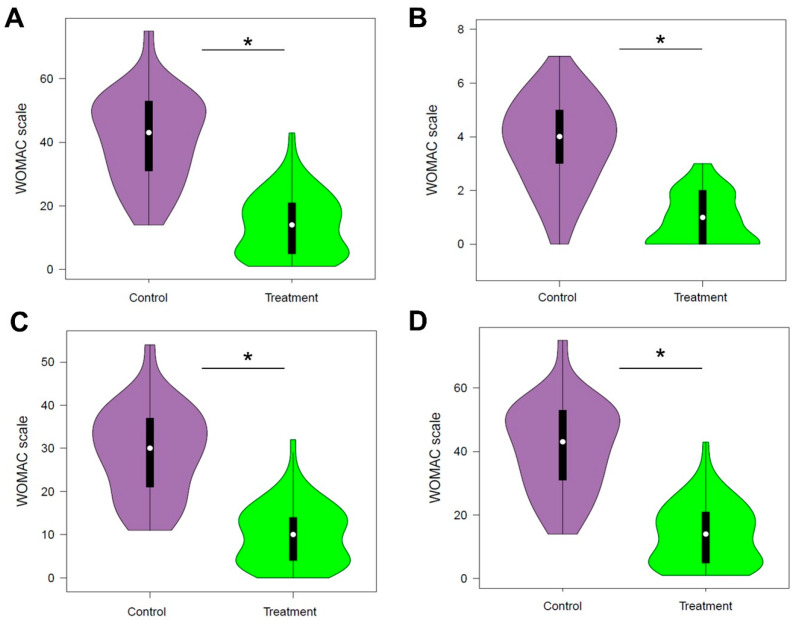
Clinical parameters evaluated in elderly individuals with osteoarthritis from the physiotherapy treatment group (green violins) and the control group using NSAIDs (purple violins). (**A**) Pain. (**B**) Stiffness. (**C**) Functional capacity. (**D**) Quality of life. * *p* < 0.001 (significant *p* < 0.05).

**Figure 3 ijerph-22-00966-f003:**
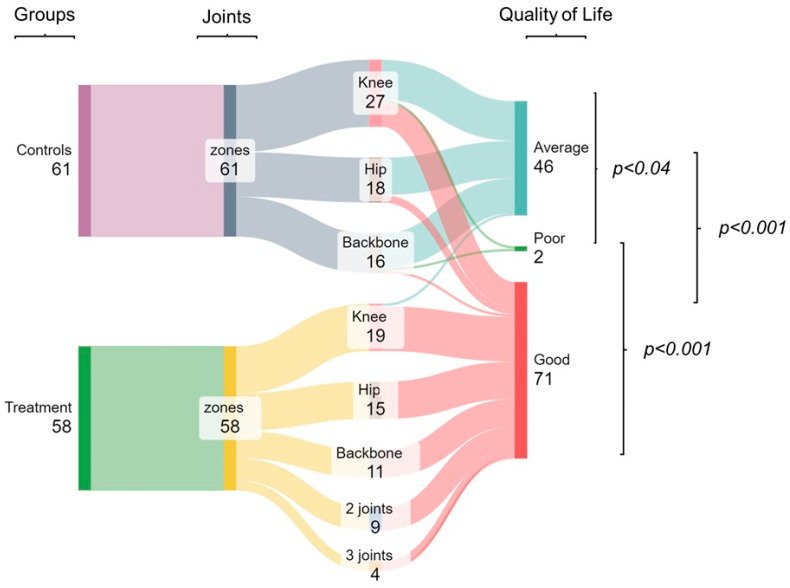
Quality of life according to the affected area with osteoarthritis in elderly individuals from the physiotherapy treatment group versus the control group. Data in N, *p* < 0.05 (significant).

**Table 1 ijerph-22-00966-t001:** Baseline characteristics of the elderly individuals included in the study with treatment and controls. Data in N (%).

Variables	Categories	Control Group	Treatment Group	Total	*p*-Value
Gender	Male	30 (61.2)	19 (38.8)	49 (41.2)	0.058
	Female	31 (44.3)	39 (55.7)	70 (58.8)	
Age range (years)	60 to 69	41 (62.1)	25 (37.9)	66 (55.5)	0.113
	70 to 79	16 (39.0)	25 (61.0)	41 (34.5)	
	80 to 89	2 (20)	8 (80)	10 (8.4)	
	90 or more	2 (100)	0 (0)	2 (1.7)	
BMI	Normal (24 to 27)	35 (44.9)	43 (55.1)	78 (65.5)	0.007
	Overweight (28 to 32)	24 (64.9)	13 (35.1)	37 (31.1)	
	Obesity (>32)	2 (50)	2 (50)	4 (3.4)	
Osteoarthritis	Knee	27 (58.7)	19 (41.3)	46 (38.7)	0.725
	Hip	18 (54.5)	15 (45.5)	33 (27.7)	
	Spine	16 (59.3)	11 (40.7)	27 (22.7)	
	two joints	0 (0)	9 (100)	9 (7.6)	
	three joints	0 (0)	4 (100)	4 (3.4)	
Background	Medication use	61 (80.3)	15 (19.7)	76 (100)	0.522

**Table 2 ijerph-22-00966-t002:** Clinical outcomes of osteoarthritis in the treatment group and the control group. Data in N (%).

	Pain			Stiffness			Functional Capacity			Total
	Low	Moderate	Severe	Low	Moderate	Severe	Adequate	With Difficulty	Severe Difficulty	
Control group	22 (36.1)	37 (60.7)	2 (3.3)	14 (23.0)	38 (62.3)	9 (14.8)	17 (27.9)	42 (68.9)	2 (3.3)	61 (100)
Intervention group	54 (93.1)	4 (6.9)	0 (0)	55 (94.8)	3 (5.2)	0 (0)	56 (96.6)	2 (3.4)	0 (0)	58 (100)
Total	76 (63.9)	41 (34.5)	2 (1.7)	69 (58.0)	41 (34.5)	9 (7.6)	73 (61.3)	44 (37.0)	2 (1.7)	119 (100)

## Data Availability

The data that have been used are confidential. Further inquiries can be directed to the corresponding author.

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
