# Peer review of "Physiotherapy Intervention Improves Clinical Outcomes and Quality of Life in Elderly Patients with Osteoarthritis: A Prospective Cohort Study"

_ijerph, 2025, doi:10.3390/ijerph22060966_

Round 1
Reviewer 1 Report (Previous Reviewer 2)
Comments and Suggestions for Authors
Thank you for your valid comments.
Author Response
The authors welcome your comments.
Reviewer 2 Report (Previous Reviewer 3)
Comments and Suggestions for Authors
The author responded to my comments.
Author Response
The authors welcome your comments.
Reviewer 3 Report (Previous Reviewer 4)
Comments and Suggestions for Authors
Thank you for taking my comments into account and making corrections. Complementing the methodology is very important. I still maintain that in order to confirm the effectiveness of a treatment method, the state before and after its use should be compared. Alternatively, the results of physiotherapy in the group also treated pharmacologically can be compared with the group treated only with NSAIDs. This raises the question of whether the physiotherapy group also received NSAIDs. All references should be described in English.
Author Response
Comment 1: Thank you for taking my comments into account and making corrections. Complementing the methodology is very important. I still maintain that in order to confirm the effectiveness of a treatment method, the state before and after its use should be compared. Alternatively, the results of physiotherapy in the group also treated pharmacologically can be compared with the group treated only with NSAIDs. This raises the question of whether the physiotherapy group also received NSAIDs.
RESPONSE 1: We agree that conducting a pre- and post-assessment could confirm the results found; however, this study identified the outcomes at the end of the intervention. The intervention group did not receive simultaneous pharmacological treatment, only the physical therapy program. As we mentioned in the manuscript, the patients had been taking medications for a long time, but they were not administered any during the intervention.
Comment 2: All references should be described in English.
RESPONSE 2: We welcome your comments, but the references have been included in the original language in accordance with the review guidelines.
Reviewer 4 Report (New Reviewer)
Comments and Suggestions for Authors
The revised manuscript is improved; however some points should be clarified.
According to Figure 1, total population is 189 and then 41 included. Was it correct? Also, patients included (n=148); however 27 were excluded. Please provide the reason why they excluded and 119 participants completed treatment. Can you provide the number of intervention and control groups? How did you random into two group? Are there anyone drop out from the study? please provide the details.
Data analysis: did you use only paired T-test? why did you not compare between group? Please recheck the statistical analysis.
Results: the average BMI in control group is higher than the intervention group. This is the reason why participants who had a high BMI is selected into the control group, what do you think? it may be selection bias.
Limitation: I am not sure, is this an observational study? if yes, the research design should be randomized control trial. If the study is experimental, the authors should provide more details regarding the methodology.
Author Response
Comment 1: According to Figure 1, total population is 189 and then 41 included. Was it correct? Also, patients included (n=148); however 27 were excluded. Please provide the reason why they excluded and 119 participants completed treatment. Can you provide the number of intervention and control groups? How did you random into two group? Are there anyone drop out from the study? please provide the details.
RESPONSE 1: A total of 189 participants were included, and those who did not meet the inclusion criteria were eliminated (n=41). Of these participants (n=148), 27 were excluded. The reasons for exclusion are described in the Methodology section, page 3, lines 105-109. And yes, 119 patients completed the treatment (intervention group 113 as it says on page 3 line 113, and 61 controls as it says on page 3 line 124). Information on study selection and withdrawal has been added.
Comment 2: Data analysis: did you use only paired T-test? why did you not compare between group? Please recheck the statistical analysis.
RESPONSE 2: We have used to compare demographic data in Table 1, to compare the outcomes of the study in Figure 1 and also the quality of life in Figure 2.
Comment 3: Results: the average BMI in control group is higher than the intervention group. This is the reason why participants who had a high BMI is selected into the control group, what do you think? it may be selection bias.
RESPONSE 3: The selection process was intentional, and despite variations in BMI scores, there was no intention to favor any specific group or treatment. As noted in the limitations, this may introduce bias; however, BMI averages were substantial in each group. Given that the entire YUYAQ household population was included, BMI selection was not possible; it's crucial to distinguish between groups in future research.
Comment 4: Limitation: I am not sure, is this an observational study? if yes, the research design should be randomized control trial. If the study is experimental, the authors should provide more details regarding the methodology.
RESPONSE 4: This research is observational because we did not completely modify the study variables; we assessed them after the physical therapy intervention and noted changes. This design is consistent with the comments of the other reviewers.
Round 2
Reviewer 4 Report (New Reviewer)
Comments and Suggestions for Authors
The revised manuscript is accepted.
This manuscript is a resubmission of an earlier submission. The following is a list of the peer review reports and author responses from that submission.
Round 1
Reviewer 1 Report
Comments and Suggestions for Authors
First of all, I would like to thank all the authors for their efforts.
Here, the authors present data indicating that physiotherapy improves clinical outcomes and quality of life in elderly patients with osteoarthritis
‘’According to the World Health Organization (WHO), osteoarthritis or arthritis is the most common disease among older adults, affecting 528 million people worldwide by 2019’’ please specify the reference
The introduction section contains very intensive information sharing. This is not good for readers. Please simplify the introduction section
At the end of the introduction section, the authors should clearly state their hypotheses
Did you perform any power analysis to determine the number of patients in the study?
‘’recent surgical history’’ is this only about knee-area surgeries or all surgeries. You should be more specific
How did you apply the questionnaires you used to the patients? In the original language or in your own language? If you used them in your own language, is there a reliability and validity study for these questionnaires?
Please explain the treatment approaches applied to the intervention group and the control group in detail under a separate heading
The discussion section should begin by stating the most important finding of the study
Comments on the Quality of English LanguageI am not able to assess that.
Author Response
RESPONSE TO REVIEWER 1
‘’According to the World Health Organization (WHO), osteoarthritis or arthritis is the most common disease among older adults, affecting 528 million people worldwide by 2019’’ please specify the reference
RESPONSE: It has been cited in the first paragraph of the introduction.
The introduction section contains very intensive information sharing. This is not good for readers. Please simplify the introduction section
RESPONSE: We have revised and improved the discussion of the study.
At the end of the introduction section, the authors should clearly state their hypotheses
RESPONSE: This information has been included in the text.
Did you perform any power analysis to determine the number of patients in the study?
RESPONSE: No. Since the entire elderly population has been included, the total population = sample has been considered.
‘’recent surgical history’’ is this only about knee-area surgeries or all surgeries. You should be more specific
RESPONSE: This information has been included.
How did you apply the questionnaires you used to the patients? In the original language or in your own language? If you used them in your own language, is there a reliability and validity study for these questionnaires?
RESPONSE: We added information about this after the paragraph explaining the validity studies the questionnaire has undergone: “Its adaptation to Spanish was carried out in 1999 by Batlle-Gualda et al. [21], and it was validated in the Peruvian population the same year by Glave-Testino et al. [22] at the Universidad Nacional Mayor de San Marcos.”
Please explain the treatment approaches applied to the intervention group and the control group in detail under a separate heading
RESPONSE: The intervention program used and control group detail has been described.
The discussion section should begin by stating the most important finding of the study
RESPONSE: We have modified the first paragraph of the discussion as suggested. Please review.
Reviewer 2 Report
Comments and Suggestions for Authors
I would like to congratulate all this for taking up this interesting study but there a major issues that need to be addressed.
1.LINES 283-286 SAYS - First, due to virtuality, there was some difficulty in having older adults respond to surveys without the guidance of the researcher or other support [36]. Therefore, we decided to use only physical surveys in face-to-face workshops and to expand the venues of the YUYAQ program by collecting information at various locations (municipality of Independencia, San Juan de Lurigancho, and San Juan de Miraflores).
I am afraid including elderly or those who could not independently respond to surveys and those who need help from the researchers to fill-in the survey forms can adversely affect the results due to this bias involved. Hence I am of the view the study needs redesigning.
2.Lines 287-292 suggest Second, it is important to consider that other factors may contribute to the beneficial effects of the intervention.For example, participants in the intervention group are part of an active aging program that includes regular physical activity workshops [17]. Some studies suggest that the I'mplementation of yoga programs can improve the effectiveness of conventional physiotherapy programs in these patients [31].
I am of the view this is a very serious limiting factor of the study. Is being stated that the participants in the intervention group are already part of an active agent problem that includes regular physical activities. Whereas the control group didn't had any treatment until the time they were inducted into study. Go for such a scenario where participants in the intervention group should have been subjected to scoring before the start of the study and at the end of the study.
3. Lines 295-297 state- Finally the emotional state of the patients was not evaluated, despite its impact on quality of life, even in those with good physical condition [32].
It goes without saying that elderly people often suffers from mental / emotional disturbances and therefore conducting a study without assessing the mental status do not dojustice to the study this is to be included in the study.
4. No objective assessment tools to assess the PROM before and after treatment was included was included in the study and hence the results of the study may not truly reflect the real values.
I urge the authors to give satisfactory explanation for the above questions.
Thank you
Author Response
RESPONSE TO REVIEWER
1.LINES 283-286 SAYS - First, due to virtuality, there was some difficulty in having older adults respond to surveys without the guidance of the researcher or other support [36]. Therefore, we decided to use only physical surveys in face-to-face workshops and to expand the venues of the YUYAQ program by collecting information at various locations (municipality of Independencia, San Juan de Lurigancho, and San Juan de Miraflores). I am afraid including elderly or those who could not independently respond to surveys and those who need help from the researchers to fill-in the survey forms can adversely affect the results due to this bias involved. Hence I am of the view the study needs redesigning.
RESPONSE: There appears to be some confusion. The researchers visited participants to supervise the completion of the surveys and clinical evaluations. We did not complete or assist in completing the questionnaires. All questionnaires were completed independently, with assistance from YUYAQ technicians and the elderly patients' relatives, precisely to avoid bias. This text has been adjusted to avoid misreading errors.
2.Lines 287-292 suggest Second, it is important to consider that other factors may contribute to the beneficial effects of the intervention.For example, participants in the intervention group are part of an active aging program that includes regular physical activity workshops [17]. Some studies suggest that the I'mplementation of yoga programs can improve the effectiveness of conventional physiotherapy programs in these patients [31]. I am of the view this is a very serious limiting factor of the study. Is being stated that the participants in the intervention group are already part of an active agent problem that includes regular physical activities. Whereas the control group didn't had any treatment until the time they were inducted into study. Go for such a scenario where participants in the intervention group should have been subjected to scoring before the start of the study and at the end of the study.
RESPONSE: We agree that this is an important factor in the study, and that's why we acknowledge it in the limitations section. There was an error in mentioning that only the intervention group participated. Government regulations exist for a physical activity program for the elderly, from which everyone has been excluded. However, since it is voluntary and often carried out in the homes of the elderly or before they are admitted to the YUYAQ, we do not have any data on this. The text has been modified to avoid errors.
- Lines 295-297 state- Finally the emotional state of the patients was not evaluated, despite its impact on quality of life, even in those with good physical condition [32]. It goes without saying that elderly people often suffers from mental / emotional disturbances and therefore conducting a study without assessing the mental status do not dojustice to the study this is to be included in the study.
RESPONSE: We agree with the reviewer regarding mental health, but it was not included in the study. This is a significant limitation, and we acknowledge and declare it. There was no funding or specialist on the team for mental health assessment, and therefore it was not assessed. It is important to consider this limitation in future studies on the topic.
- No objective assessment tools to assess the PROM before and after treatment was included was included in the study and hence the results of the study may not truly reflect the real values.
RESPONSE: The study design focused on comparing clinical improvement outcomes between the two groups, not on the final outcome. We understand that pre- and post-treatment evaluations are important; however, this study focused on evaluating two treatments at the same time in two groups of elderly patients.
Reviewer 3 Report
Comments and Suggestions for Authors
- The author needs to updated the statistical epidemiology information.
For instance, According to the World Health Organization (WHO), osteoarthritis or arthritis is the most common disease among older adults, affecting 528 million people worldwide by 2019. Need to be updated by 2025.
- Non-Randomized Cohorts: Lack of randomization introduces selection bias. Baseline BMI differences (26.1 vs. 27.6, p=0.007) may confound results.
- Unblinded Intervention: Participants/assessors aware of group allocation risk performance/reporting bias.
- The inclusion and exclusion criterial is not clear.
- Physiotherapy program included yoga, tai chi, and cardio dance. Combined modalities preclude identification of the most effective component(s).
- The Physiotherapy program needs to explain in detail.
- NSAID regimen (types, dosages, adherence) not specified. Variability in pharmacological management undermines comparability.
- Missing Covariate Adjustment: Significant baseline BMI differences were not adjusted for in analyses, potentially inflating treatment effect estimates.
Author Response
RESPONSE TO REVIEWER 3
The author needs to updated the statistical epidemiology information. For instance, According to the World Health Organization (WHO), osteoarthritis or arthritis is the most common disease among older adults, affecting 528 million people worldwide by 2019. Need to be updated by 2025.
- RESPONSE: The WHO has considered this report to be the latest on osteoarthritis, which is why we cite it in the article as official information.
Non-Randomized Cohorts: Lack of randomization introduces selection bias. Baseline BMI differences (26.1 vs. 27.6, p=0.007) may confound results.
RESPONSE: Randomization was not performed because a probability sample was not used. Exclusion criteria were met from the total population, and the two groups were standardized based on patient availability. Nonsignificant differences were observed in almost all demographic variables, with the exception of BMI, although the standard deviations were correlated and overlapped.
Unblinded Intervention: Participants/assessors aware of group allocation risk performance/reporting bias.
RESPONSE: Yes, the authors were aware of the group assignment, but we maintained impartiality. The participants, as were the participants' employees and family members, were blinded to their group assignment.
The inclusion and exclusion criterial is not clear.
RESPONSE: This section has been improved
Physiotherapy program included yoga, tai chi, and cardio dance. Combined modalities preclude identification of the most effective component(s).
RESPONSE: The intervention program used has been described.
The Physiotherapy program needs to explain in detail.
RESPONSE: The intervention program used has been described.
NSAID regimen (types, dosages, adherence) not specified. Variability in pharmacological management undermines comparability.
RESPONSE: This section has been improved, we added information of pharmacological treatment.
Missing Covariate Adjustment: Significant baseline BMI differences were not adjusted for in analyses, potentially inflating treatment effect estimates.
RESPONSE: We disagree with the reviewer on this point. Although the BMIs differed between the groups, the means and SDs were close. A study needs to have everything paired, but in this case, we made sure to control for most of the variables.
Reviewer 4 Report
Comments and Suggestions for Authors
The article deals with elderly patients with osteoarthritis. This is a current and important topic. In the treatment of this disease, methods of treatment that reduce pain, delay the progression of changes and improve functionality are important, thus delaying surgical intervention. The aim of this work was to "compare the effects of a physiotherapeutic intervention on pain, stiffness, functional capacity, and perceived quality of life”. The study's methodology did not explain what physiotherapy intervention meant or what the physiotherapy treatment consisted of in the intervention group. Physical activity such as yoga, dance, tai chi is not the same as physiotherapy treatment, which includes physical therapy, massage, manual therapy and, above all, exercises adapted to the condition of the disease. The qualification of patients to the study groups is also questionable. What degree of advancement of degenerative changes and what degree of pain determined the qualification of patients for the study? This is particularly important in functional assessment. Were NSAIDs administered to patients with mild pain? There is also no information about the treatment in the control group regarding the treatment regimen and duration of pharmacotherapy. A well-known WOMAC questionnaire was used for the study to assess symptoms (pain with activities and stiffness) and functional assessment in knee and hip osteoarthritis. Other questionnaires, such as the SF-36, are worth using to assess quality of life. Another doubt is raised by the way the study was conducted "at the end of the phsysiotherapy intervention and follow-up period for each study group”. To demonstrate the effectiveness of a treatment method, it is necessary to compare the condition before and after the use of a given treatment method, and in the same way to compare the results when using another method. The results were analyzed in detail. However, it is difficult to agree with the conclusions if the results of the questionnaire before undertaking physical activity and before starting pharmacotherapy are not presented. There is no reference to the second reference in the text. Access to references 5 and 17 was not obtained.
Author Response
REVIEWER 4
The article deals with elderly patients with osteoarthritis. This is a current and important topic. In the treatment of this disease, methods of treatment that reduce pain, delay the progression of changes, and improve functionality are important, thus delaying surgical intervention. The aim of this work was to "compare the effects of a physiotherapeutic intervention on pain, stiffness, functional capacity, and perceived quality of life”.
The study's methodology did not explain what physiotherapy intervention meant or what the physiotherapy treatment consisted of in the intervention group. Physical activity such as yoga, dance, tai chi is not the same as physiotherapy treatment, which includes physical therapy, massage, manual therapy, and, above all, exercises adapted to the condition of the disease.
RESPONSE: The intervention program used has been described.
The qualification of patients for the study groups is also questionable.
RESPONSE: We have explained and improved this section.
What degree of advancement of degenerative changes and what degree of pain determined the qualification of patients for the study? This is particularly important in functional assessment. Were NSAIDs administered to patients with mild pain? There is also no information about the treatment in the control group regarding the treatment regimen and duration of pharmacotherapy.
RESPONSE: The control group has been described, and we added information of the treatment.
A well-known WOMAC questionnaire was used for the study to assess symptoms (pain with activities and stiffness) and functional assessment in knee and hip osteoarthritis. Other questionnaires, such as the SF-36, are worth using to assess quality of life.
RESPONSE: We agree with the reviewer regarding other instruments for assessing quality of life; however, to simplify the survey time per patient and given the WOMAC quality of life medicine values, we used a single questionnaire.
Another doubt is raised by the way the study was conducted, "at the end of the physiotherapy intervention and follow-up period for each study group”. To demonstrate the effectiveness of a treatment method, it is necessary to compare the condition before and after the use of a given treatment method, and in the same way to compare the results when using another method. The results were analyzed in detail. However, it is difficult to agree with the conclusions if the results of the questionnaire before undertaking physical activity and before starting pharmacotherapy are not presented.
RESPONSE: We agree with the reviewer on the importance of conducting a pre- and post-intervention evaluation. However, in this study, we focused on a general comparison of interventions, as a first approximation at the YAYAQ center. Future studies with longer follow-up periods and controlling for more variables may be needed to determine the true effect of physical therapy. Although limited by this weakness, this research paves the way for evaluations focused on physical therapy as a replacement for anti-inflammatory drugs in Peru. We have added this limitation.
There is no reference to the second reference in the text. Access to references 5 and 17 was not obtained.
RESPONSE: We have reviewed the references and adjusted everything